# Investigating the Impact of Electrostatic Interactions on Calmodulin Binding and Ca^2+^-Dependent Activation of the Calcium-Gated Potassium SK4 Channel

**DOI:** 10.3390/ijms25084255

**Published:** 2024-04-11

**Authors:** Émilie Segura, Juan Zhao, Marlena Broszczak, Frédéric Audet, Rémy Sauvé, Lucie Parent

**Affiliations:** 1Département de Pharmacologie et Physiologie, Faculté de Médecine, Centre de Recherche de l’Institut de Cardiologie de Montréal, Université de Montréal, Montréal, QC H1T 1C8, Canada; emilie.segura@umontreal.ca (É.S.); frederic.audet.1@umontreal.ca (F.A.); 2Centre de Recherche de l’Institut de Cardiologie de Montréal, Université de Montréal, Montréal, QC H1T 1C8, Canada; juan.zhao@icm-mhi.org (J.Z.); marlena.broszczak@icm-mhi.org (M.B.); 3Département de Pharmacologie et Physiologie, Faculté de Médecine, Université de Montréal, 2900 Bd Édouard-Montpetit, Montréal, QC H3T 1J4, Canada; remy.sauve@umontreal.ca

**Keywords:** protein–protein interaction, co-immunoprecipitation, electrophysiology, calmodulin, ion channel, calcium

## Abstract

Ca^2+^ binding to the ubiquitous Ca^2+^ sensing protein calmodulin (CaM) activates the intermediate conductance Ca^2+^-activated SK4 channel. Potential hydrophilic pockets for CaM binding have been identified at the intracellular HA and HB helices in the C-terminal of SK4 from the three published cryo-EM structures of SK4. Single charge reversal substitutions at either site, significantly weakened the pull-down of SK4 by CaM wild-type (CaM), and decreased the TRAM-34 sensitive outward K^+^ current densities in native HEK293T cells when compared with SK4 WT measured under the same conditions. Only the doubly substituted SK4 R352D/R355D (HB helix) obliterated the CaM-mediated pull-down and thwarted outward K^+^ currents. However, overexpression of CaM E84K/E87K, which had been predicted to face the arginine doublet, restored the CaM-mediated pull-down of SK4 R352D/R355D and normalized its whole-cell current density. Virtual analysis of the putative salt bridges supports a unique role for the positively charged arginine doublet at the HB helix into anchoring the interaction with the negatively charged CaM glutamate 84 and 87 CaM. Our findings underscore the unique contribution of electrostatic interactions in carrying CaM binding onto SK4 and support the role of the C-terminal HB helix to the Ca^2+^-dependent gating process.

## 1. Introduction

Ca^2+^-activated K^+^ conductance was first identified in the nerve cells from mollusks [1], but evidence for a Ca^2+^-dependent K^+^ permeability was reported in erythrocytes back in 1958 [2]. Ca^2+^-activated potassium (K^+^) channels are encoded by the KCN genes in mammals. KCNMA1 (BK channels or KCa1.1), KCNN1 (SK1 or KCa2.1), KCNN2 (SK2 or KCa2.2), KCNN3 (SK3 or KCa2.3), and KCNN4 (SK4 or IK or KCa3.1) have been identified as crucial components of numerous physiological processes in epithelia [3], in the central nervous system [4], and in the cardiovascular system [5,6,7]. In particular, SK4 channels are expressed in mammalian pacemaker and atrial cardiomyocytes [8], and their specific pharmacological inhibition paves the way for a novel therapeutic strategy in the treatment of atrial fibrillation [8].

K^+^-selective SK channels belong to the tetrameric 6-transmembrane segment cation channel family. They are chemo-mechanically gated at submicromolar concentrations of free intracellular [Ca^2+^] or Cai in a dose-dependent fashion with an EC_50_ ≈ 0.25–0.9 μM [9] through the Ca^2+^-sensing protein calmodulin (CaM) that is constitutively bound to the pore-forming subunit [10]. CaM is a flexible bi-lobed 148-amino acid protein linked by a phosphorylation-prone interlobe region (78-DTDS-81 region). Each lobe, formed by a pair of EF-hand motifs, EF1/EF2 in the N-lobe and EF3/EF4 in the C-lobe, can bind up to two Ca^2+^ ions [11,12,13,14]. Alanine substitution of the aspartate residue found in the first position of each EF hand (Asp20 in EF1, Asp56 in EF2, Asp93 in EF3 and Asp129 in EF4) abolishes Ca^2+^ binding. [15]. The four pore-forming α subunits of the tetrameric SK4 channel are each constitutively associated with one CaM protein in a 4:4 arrangement [16]. The intracellular C-terminal of each pore-forming subunit is formed by three consecutive and anti-parallel helices, termed HA (Lys293-Arg330), HB (Ser334-Met368), and HC (Ile371-Leu385) [16]. The HA and HB helices almost run parallel to the membrane plane between two SK4 opposite subunits (Figure 1A,B). The C-lobe of CaM, remains lodged within the fold of the HA and HB helices, after the addition of Ca^2+^ (6CNN.pdb and 6CNO.pdb structures) [17], leading to the conclusion that this region is not contributing to the Ca^2+^-dependent gating process. 

In contrast, the N-lobe of CaM experienced large conformational changes when comparing its structural coordinates between the Ca^2+^-free and the Ca^2+^-loaded forms of the channel structures [16,17] leading to the postulate that Ca^2+^-dependent gating was controlled by the binding of the N-lobe of CaM onto the intracellular S_4_–S_5_ helices of SK4. The molecular events occurring at the C-terminal helices of SK4 initiated upon CaM binding remains to be documented, in particular their impact on the activation of SK4. In addition, the molecular mechanisms responsible for CaM binding are not fully understood. Close inspection of the contact surface area between the SK4 α subunit and the C-lobe of CaM revealed a significant contribution from hydrophobic and/or aromatic residues (Leu319, Trp323, and Tyr326 in SK4; Met76, Val91, Phe92, Leu112, Met144, Met145, and Ala148 in CaM), as well as from polar and/or charged residues (Tyr320, Asn348, Arg352, and Arg355 in SK4; Asp78, Asp84, and Glu127 in CaM). It has been established that short-range hydrophobic interactions account for the interaction of the Ca^2+^-loaded holo-CaM form with the IQ domain in the C-terminal of Ca_V_1.2 [18], and for anchoring CaM binding to the HA helix in the 3-D model of the SK2 channel [19]. In contrast, a doublet of positively charged residues at the C-terminal of SK2 was shown to play a role in mediating channel function and surface trafficking [20]. It is not clear what mechanism is dominant in SK4 given the differences in the Cai sensitivity of SK2 and SK4 [9,21]. 

It is generally agreed that Ca^2+^-free CaM (apo-CaM) binding involves predominantly charged residues [22]. We thus postulated that long-range electrostatic interactions are responsible for the constitutive binding of CaM to the inner HA and HB helices of SK4 based upon the analysis of the cryo-EM structure [16]. To correlate CaM binding and channel activation, our approach combined a pull-down assay based on co-immunoprecipitation of the full-length SK4 and CaM (WT and substituted proteins) carried out in micromolar contaminating Ca^2+^ solutions (no Ca^2+^ chelator), compared with functional assays performed using whole-cell patch-clamp recordings of the same pair of proteins in the presence of 0.1 µM Cai. CaM binding and channel function were eliminated after the double charge substitution at positions SK4 Arg352 and Arg355 (R352D/R355D). Whole-cell currents were not recovered after overexpression of recombinant CaM WT but were fully rescued at submicromolar 0.03 µM Cai after the co-expression of the SK4 R352D/R355D with CaM E84K/E87K, which involved a complete reciprocal charge inversion on both proteins. Furthermore, although the CaM E84K/E87K-mediated pull-down signal for SK4 R352D and SK4 R355D were qualitatively equivalent, the Ca^2+^-sensitivities of their outward K^+^ current densities differed markedly. These observations argue that molecular events occurring at the C-terminal of SK4 influence its Ca^2+^-dependent activation gating. We show that electrostatic interactions between the CaM C-lobe and the HB helix of SK4 account for CaM binding while modulating the Ca^2+^ sensitivity of the channel gating process.

## 2. Results

### 2.1. Two Hydrophilic Pockets Are Predicted between CaM and SK4 HA-HB Helices

Structural predictions were extracted from the 6CNO.pdb cryo-EM structure [16] that captures the channel with CaM in the Ca^2+^ bound state II. A virtual analysis of hydrophilic interactions suggests the presence of two hydrophilic pockets. The first hydrophilic pocket entails interactions between EF4-CaM and the HA helix of SK4, namely CaM Glu123 and SK4 Lys327 and between CaM Glu127 and SK4 Lys327 with estimated average O-N distances of 5.2 ± 0.9 Å and 4.5 ± 0.2 Å, respectively (Figure 1C). The second pocket is formed by negatively charged glutamate residues in EF3-CaM and positively charged arginine residues in the HB helix of SK4. Pairs formed by CaM Glu84 and SK4 Arg352; CaM Glu84 and SK4 Arg355; CaM Glu87 and SK4 Arg355 are compatible with the presence of salt bridges (Figure 1D). Estimated O-N distances averaged 4.1 ± 0.5 Å, 4.4 ± 0.7 Å, and 5.5 ± 0.5 Å respectively over a 1.6-µs simulation time period. Pairs of residues with a cut-off average distance smaller than 6 Å were investigated after heterologous expression of the charge substituted proteins. This analysis supports the proposition that charged residues from SK4 and CaM are involved in the binding of the C-lobe of CaM to SK4.

### 2.2. Endogenous CaM Supports Function of Recombinant SK4 in HEK293T Cells

The role of electrostatic interactions was investigated with SK4 and CaM constructs engineered to reverse the net charge carried by the target residue. SK4 was successfully expressed in recombinant HEK293T cells in the presence of endogenous CaM and in the presence of recombinant CaM (Figure 2). Robust voltage-independent K^+^ currents were measured in the whole-cell configuration with 0.1 µM free [Ca^2+^] or Cai in the patch pipette under a [K^+^] gradient of 140 mM (inside)/3 mM (out). Whole-cell currents were recorded using square voltage pulse (Figure 2A,C) or voltage ramps (Figure 2B,D). In every case, the resulting current/voltage curves appeared linear over the −120 to +100 mV range, with a reversal potential estimated at −80 mV. Total currents were >95% inhibited by the bath application of 0.3 μM TRAM-34 (Figure 2). Only TRAM-34 sensitive whole-cell K^+^ currents measured at +60 mV are reported in Tables. The TRAM-34 insensitive currents displayed similar current densities and voltage-dependence as the outwardly rectifying endogenous K^+^ currents recorded in mock-transfected cells with current densities ranging from 50 to 70 pA/pF at +60 mV (Table 1). No TRAM-34 sensitive currents could be recorded in the presence of 0.01 μM Cai, whereas 0.03 to 0.1 µM Cai generated robust channel function (>500 pA/pF at +60 mV). We opted to work under conditions where the Ca^2+^ concentration in the patch pipette is close to the EC_50_ for Ca^2+^ ions (EC_50_ ≈ 95 nM) [21]. Co-transfection with pIRES-DsRed-HisB-cMyc-CaM WT—the construct used for the protein assays—did not further increase the whole-cell current density under any Cai tested, indicating that endogenous CaM is not rate limiting in native HEK293T cells [8,23]. 

### 2.3. Single Substitution at Lys327 in the SK4 HA Helix Decrease Interaction with CaM

The relative role of the hydrophilic pocket between negatively charged glutamate residues Glu123 and Glu127 in CaM EF-4 and the positively charged residue Lys327 within the SK4 HA helix (Figure 1C) was evaluated using co-immunoprecipitation assays. The latter are useful to detect high-affinity interactions but could only provide a qualitative estimation of protein interaction relative to the signal recorded from other constructs. SK4 WT was successfully pulled down by pIRES-DsRed-HisB-cMyc-CaM WT by anti-His beads capturing His-tagged CaM proteins (Figure 3(Aii)). The presence of recombinant CaM proteins after elution of the beads was confirmed by the protein signal captured by the anti-His antibody that appeared at approximately 20 kDa. SK4 WT pull-down was not impaired by the single charge substituted CaM E123K and E127K, the double CaM E123A/E127A, or the double CaM E123K/E127K proteins, indicating that the negative charge at these positions in the EF-hand 4 of CaM do not contribute significantly to interaction with SK4 WT in low Ca^2+^ conditions. Nonetheless co-immunoprecipitation of SK4 K327E with CaM WT was disrupted and the destabilized interaction of SK4 K327E was partially reversed with CaM E123K/E127K or CaM E123A/E127A supporting a potential interaction between the HA helix of SK4 and CaM EF-4 (Figure 3A,B). The strong pull-down signal for SK4 K327E and CaM E123A/E127A suggests that the interaction might not primarily arise from the formation of salt bridges. 

Whole-cell K^+^ outward currents were significantly lower for SK4 K327E in the presence of endogenous CaM (Table 1) whereas overexpression of the recombinant CaM WT construct rescued its function to the level of SK4 WT with 405 ± 193 pA/pF (Table 1). Therefore, SK4 K327E weakened the binding of CaM onto the HA helix of SK4, although SK4 function was normalized in the presence of a higher cellular concentration of CaM WT. Increasing the cellular CaM concentration can thus overcome the weakened interactions by increasing the probability of CaM occupying the binding site. This in turn should result in a higher channel activation rate and cause a decrease in the apparent Ca^2+^ sensitivity of the gating process enabling channel activation at 0.1 μM Cai. Not surprisingly, outward K^+^ currents, measured under the same experimental conditions, were similar for SK4 K327E with CaM WT or with CaM E123K/E127K with 382 ± 182 pA/pF (N = 1, n = 7) for the latter combination. 

### 2.4. Double Charge Substitution in the SK4 HB Helix Disrupts Protein Interaction and Channel Function

Charge substitution of single arginine residues at position Arg352 or Arg355 at the HB helix (Figure 1D) decreased the pull-down signal of the SK4 protein (Figure 4(Aii)). However, the double charge substitution annihilated the signal with relative intensity diminishing from SK4 WT > R352D >> R355D > R352D/R355D, while the total protein expression in the input fraction was equally detected in all SK4 constructs (Figure 4(Ai), see also Figure 3, Figure 5, and Figure 6). The absence of pull-down signal for the doubly substituted SK4 R352D/R355D argues that the doublet of positively charged residues plays a unique role in carrying CaM binding onto SK4 under the prevailing submillimolar Ca^2+^ conditions of our pull-down assays.

Patch-clamp experiments carried out in the presence of 0.1 µM Cai and endogenous CaM mirrored the pull-down data with outward current densities decreasing from SK4 WT > R352D >> R355D ≈ R352D/R355D ≈ R352D/R355D/K327E (Figure 4B). Current densities for SK4 R355D, R352D/R355D, and R352D/R355D/K327E were not significantly different from endogenous K^+^ control currents. Note that endogenous CaM appeared to be sufficient to traffic the single charge- substituted SK4 R352D and R355D proteins at the cell surface; therefore, the lack of function was not solely caused by impaired transport to the cell membrane (Figure 5). Overexpression with the recombinant CaM WT construct rescued the function of SK4 R352D and R355D to the level of SK4 WT but not the function of SK4 R352D/R355D and SK4 R352D/R355D/K327E (Figure 4C, Table 1). SK4 Arg352 and Arg355 appear to contribute cooperatively to the binding of CaM and to channel function. 

### 2.5. Electrostatic Interactions Promote CaM Binding onto the SK4 HB Helix

In the 3-D structure, negatively charged CaM residues Glu84 and Glu87 are predicted to be facing the positively charged Arginine residues in the HB helix of SK4. Furthermore, our analysis of contact surface areas predicted that Glu84 exhibits the most extensive contact area with SK4 within the CaM C-lobe among other CaM residues. The cDNA coding for SK4 WT and substituted R352D, R355D, R352D/R355D were overexpressed with substituted CaM E84K (Figure 6A) or E87K (Figure 6B), respectively. We reasoned that if a single electrostatic interaction was necessary and sufficient, CaM binding would be restored by pairing the charge reversal in both SK4 and CaM. Pull-down experiments clearly show that protein interaction of SK4 WT was impaired with CaM E84K indicating that CaM Glu84 contributes to CaM binding onto SK4 WT. 

The protein signal for SK4 R352D was relatively stronger than the signal for R355D or R352D/R355D suggesting that protein interaction was somewhat recovered by the paired charge switch between SK4 and CaM. In agreement, CaM E84K partially restored the protein interaction with SK4 R352D/R355D, an interaction that was not observed with CaM WT. By comparison, protein signal was nearly absent with the pair SK4 R355D + CaM E84K arguing that CaM E84K is unable to promote significant interaction with SK4 R355D. In contrast to CaM E84K, co-expression with CaM E87K did not disrupt the interaction with SK4 WT (Figure 6B). The pull-down signal intensity was however similarly and greatly decreased for the three SK4 constructs with SK4 WT >> R352D ≈ R355D ≈ R352D/R355D. CaM E87K alone did not affect the interaction of CaM with SK4 WT; this is the only charge substitution in CaM that prevented the interaction of SK4 R352D. In this case, we cannot exclude the possibility of an indirect effect stemming from E87K interacting with CaM Glu84, potentially disrupting the interaction between CaM Glu84 and SK4 Arg352. MD simulations suggest that the repulsion exerted by the presence of positively residues CaM E87K and SK4 Arg355 facing each other could interfere to CaM binding in the SK4 R352D + CaM E87K pair. Indeed, CaM E84K/E87K restored the pull-down signal for SK4 R352D and SK4 R352D/R355D (Figure 7A). The strong signal observed for SK4 R352D/R355D with CaM E84K/E87K anti-His beads, confirmed in three separate experiments carried out in distinct transfections, argues that CaM Glu84 and to a lesser extent CaM Glu87 are involved in the constitutive CaM binding onto SK4 and further suggests that together with SK4 Arg352 and Arg355 they form a minimum electrostatic «quadrinome».

The quadruple charge reversal also fully restored channel function of SK4 R352D/R355D. Whole-cell TRAM-34 sensitive outward K^+^ currents, recorded in the presence of 0.1 µM Cai, were larger for SK4 R352D, R355D, and R352D/R355D with CaM E84K/E87K than with endogenous CaM or CaM WT (Figure 7B, Table 2). In contrast, SK4 K327E yielded the same current density in the presence of CaM E84K/E87K and in the presence of endogenous CaM indicating that the CaM construct did not supersede endogenous CaM at the binding site. Under our experimental conditions, endogenous CaM and overexpressed recombinant CaM will competition for the same CaM binding site (CaMBD), with the higher affinity and/or more abundant CaM form being more likely to occupy the CaMBD. The larger currents recorded for R325D, R355D, and R352D/R355D with CaM E84K/E87K than with endogenous CaM argue that CaM E84K/E87K interacts strongly with all three constructs The free (Ca^2+^) bound -CaM concentrations in HEK293T cells, estimated to be in the nanomolar range [23,24], appears to be sufficient for SK4 WT to produce similar outward current densities with either endogenous or overexpressed recombinant CaM. The one remaining unknown is the ratio of phosphorylated to non-phosphorylated CaM that is the essential parameter in the control of SK channel function [8,25,26].

In the presence of CaM E84K/E87K, SK4 R352D, R355D, and R352D/R355D generated similar whole-cell current densities when measured with 0.1 µM Cai. The similar current densities recorded for these constructs were unexpected given the sharp difference in the relative strength of their respective pull-down by CaM E84K/E87K. Channel function, however, requires the Ca^2+^-bound form of CaM with higher currents predicted as Cai increases. We reasoned that the differences in pull-down signals could be teased out by using a lower Cai when performing the functional assays. Whole-cell currents were thus recorded again with 0.03 μM free Cai in the patch pipette. As seen, channel function was significantly larger for R352D/R355D than for the single substituted SK4 proteins (Figure 7C). Notably, adding the K327E substitution located on the SK4 HA helix significantly impeded the functional recovery of SK4 RD/RD with the CaM EK/EK background, suggesting that the SK4 HA helix could be involved in channel gating. Altogether, these results show that the pair of electrostatic interactions formed between the HB helix of SK4 (Arg352 and Arg355) and the EF-3 of CaM (Glu84 and Glu87) is essential to carry CaM interaction and channel function, and further suggest that the Ca^2+^-dependent modulation of the channel open state by Cai is influenced by events occurring at the SK4 HB helix. 

## 3. Discussion

### 3.1. Electrostatic Interactions Underlie CaM Binding on the HB Helix of SK4

In contrast to the partial X-ray structures of the SK2 channel [27,28], virtual analysis of the three cryo-EM structures from SK4 (6CNO.pdb, 6CNN.pdb, and 6CNM.pdb) identified two potential hydrophilic pockets for the CaMBD in the C-terminal of SK4, formed by electrostatic interactions between positively charged guanidium groups bearing residues in SK4 and negatively charged carboxylate-bearing residues in CaM. Long-range electrostatic interactions are not usually believed to underlie strong protein binding although it is not ruled out for apo-CaM [22]. We thus undertook a detailed analysis of the predicted electrostatic network between the CaM-C lobe and the C-terminus of the SK4 channel. Herein, we have shown that the single charge substitution on the EF-3 hand of CaM at position Glu84 (E84K) but not the single substituted CaM E87K disrupted the pull-down of SK4 WT by CaM. The binding of CaM WT, CaM E84K, and CaM E87K to SK4 R355D and to R352D/R355D was impaired. Only the dual charge reversal in CaM E84K/E87K promoted CaM pull-down to SK4 R352D/R355D. In contrast to SK4 R355D, SK4 R352D was pulled down by CaM WT, CaM E84K, and CaM E84K/E87K and produced outward currents in the presence of endogenous CaM.

The greater tolerance of SK4 Arg352 versus Arg355 to charge substitution in the EF3 motif of-CaM points to Arg355 as being more sensitive to changes in its interactome at the CaMBD. This observation agrees with the report that the current densities of SK4 R352Q are equivalent to SK4 WT whereas the activity of SK4 R355G is near the resolution background level when measured under the same conditions in CHO cells [8]. Nonetheless, the stronger impact of the double charge substitution SK4 R352D/R355D when compared to R355D, abrogating interaction with CaM WT and preventing channel function with CaM WT suggest that the positively charged Arg352 and Arg355 play complementary roles role at the CaMBD interface. 

To explore the molecular events accounting for the experimental results obtained with the SK4 Arg352/Arg355 doublet, MD simulations (1.6 µs) were performed with the Ca^2+^-bound CaM structure (6CNO.pdb) and the Ca^2+^-free CaM (6CNM.pdb) 3-D models. The former is believed to account for the open state while the latter is used to represent the closed state of the channel. Our experimental conditions are using supraphysiological micromolar calcium concentration for the pull-down studies and submicromolar Cai for measuring channel function in patch-clamp recordings.

An MD analysis based upon the “open state” model suggests that SK4 Arg352 and SK4 Arg355 in the HB helix interact alternatively with either CaM Glu84 or Glu87 in the EF-3 hand of CaM. During the timeframe of the MD simulations, CaM Glu84 demonstrated similar effectiveness in interacting with both SK4 Arg352 and Arg355, whereas CaM Glu87 displayed a predominant interaction with SK4 Arg355. In this scenario, SK4 Arg352 should interact significantly less frequently and with lower intensity, with CaM Glu87 than with CaM Glu84. It ensues that the protein–protein interaction promoted by the double charge reversal in the SK4 R352D + CaM E84K would dominate over the native interaction between SK4 Arg355 and CaM Glu87 and that the protein–protein interaction promoted by the double charge reversal in the SK4 R355D + CaM E84K complex would dominate over the native interaction between SK4 Arg352 and CaM Glu87. We report that CaM E84K pulled down SK4 R352D, but we failed to observe a high-affinity interaction between CaM E84K and SK4 R355D. In addition, CaM E87K pulled down SK4 WT but neither SK4 R352D, R355D, nor R352D/R355D, which was as if the interaction between CaM E87K and either SK4 R352D or SK4 R355D was negligible. MD simulations, carried out with the “open state” 3-D model modified to include the double R352D/R355D substitution, also support the alternative binding of CaM Glu84 and Glu87 onto the doublet of Arg residues. The robust alternative binding could account for the observation that CaM WT could not pull-down SK4 R352D/R355D and that CaM E84K/E87K pulled down SK4 R352D and SK4 R352D/R355D. It is, however, incompatible with the observation that CaM E84K/E87K was unable to pull down SK4 R355D. The overall experimental behavior of SK4 R352D also markedly differed from SK4 R355D, an observation not predicted from the MD simulations carried out with the “open state” model.

In contrast, the 3-D model based upon the “closed state” 6CNM.pdb structure fails to predict any interaction between SK4 Arg355 and CaM Glu84 in the native SK4 channel complex or between SK4 R355D and CaM E84K in the SK4 R355D + CaM EK/EK complex. In addition, the “closed state” model does not support protein interaction between SK4 R352D and CaM E87K in the SK4 R352D + CaM EK/EK complex or within the SK4 R352D + CaM E87K complex. The strongest interacting pair is being formed between SK4 Arg352 and CaM Glu84. Contributions from the other pairs appear to be more modest. Thus, our co-immunoprecipitation data carried out in the absence of a divalent cation chelator, seem to be better accounted for by the “closed state” than by the “open state” model. The comparative analysis of the “open” and “closed” models further suggest that the relative strength of the protein interaction at the CaMBD located in the C-terminal of SK4 can be modulated by local Cai. 

The analysis of the predicted contact area at the CaMBD, performed with either 3-D model, supports nonetheless a complete disruption in the interaction between the HB helix of SK4 and CaM in the presence of SK4 R352D/R355D. Positively charged guanidium groups in this short hydrophilic segment, seem to be required to stabilize the CaM/SK4 interaction despite the ongoing presence of hydrophobic interactions anchored by SK4 residues Leu319, Trp323, and Try326 in the HA helix. Our analysis supports a significant role for electrostatic interactions in mediating the binding of the C-lobe of CaM onto the intracellular helices located at the C-terminal of SK4.

### 3.2. A Role for the Intracellular C-Terminal of SK4 in the Channel Activation Gating

Beyond the identification of interaction sites, the analysis of the three cryo-EM structures led to propose a novel gating mechanism [16]. In contrast to the partial X-ray structures [27,28], each SK channel subunit was reported to bind to a single CaM molecule with a maximum Ca^2+^ binding capacity of 3 Ca^2+^ ions per CaM (2 Ca^2+^ ions on the N-lobe, 1 Ca^2+^ on the C-lobe). In the presence of Ca^2+^, the unusually long cytosolic S4–S5 linker (absent from the partial X-ray structures) formed by the folding of two α-helices S_45_A and S_45_B was observed to interface with the CaM N-lobe [16]. The binding of the N-lobe of Ca^2+^-bound CaM to the S_45_A helix is predicted to ultimately open the channel gate by pulling the S6 pore helices sufficiently to let K^+^ ions out of the cell. Given that the position of the CaM C-lobe is not reported to be significantly influenced by the presence of Ca^2+^ on CaM [16,17], the activation gating of SK channels could be envisioned as a sequential process. Ca^2+^-independent binding of the C-lobe of CaM onto the C-terminal of the pore-forming α subunit is proposed as the initial step with little or no further contribution to the Ca^2+^-dependent channel opening. Recent data do not, however, completely concur with this proposition. Studies on the related SK2 channel demonstrated that the N- and C-lobes of CaM need to remain structurally connected for the channel to open in response to Ca^2+^ [29]. In addition, the function of SK2 R464E/K467E (homologous to SK4 R352D/R355D) construct was recovered by intracellular perfusion with 10 µM purified Ca^2+^-CaM WT [20]. These two findings argue that channel activation could be initiated despite weakened protein interactions with the CaM C-lobe. In fact, channel function was recorded for pairs of constructs showing no discernible interaction in supraphysiological [Ca^2+^]. Current models of the SK4 gating process propose a rotation of the C-lobe of CaM bound to the SK4 HA and/or HB helices in response to the movement of the SK4 S_45_A/S_45_B helices bolstered by Ca^2+^ binding to the CaM N-lobe. This process is expected to be modulated by the strength of the protein–protein interaction at the SK4 HA-HB helices. Weaker interactions could make successful rotations less likely, leading to a reduction in channel activation rates and, consequently, a decrease in the apparent Ca^2+^ sensitivity of the gating process. On one hand, patch-clamp experiments carried out with CaM WT borne out that prediction. Increasing Ca^2+^-CaM WT successfully rescued TRAM-34 sensitive K^+^ currents of the single-substituted SK4 K327E and R355D, with current densities equivalent to SK4 WT. On the other hand, the activation of the complex formed by SK4 R352D + CaM E84K/E87K required higher Cai when compared with the complex formed by SK4 R352D/R355D + CaM E84K/E87K, despite producing qualitatively similar protein signals in the pull-down assays. In addition, SK4 R355D + CaM E84K/E87K generated “normal” current densities at 0.1 μM Cai, despite the absence of pull-down by CaM E84K/E87K. The last two observations indicate that increasing Cai within the submicromolar range, without increasing cellular CaM, could be sufficient to offset the negative impact produced by weakened CaM binding to the full-length SK4 protein. Our results thus support a model of activation gating that is dynamically controlled by events occurring simultaneously at the intracellular S4–S5 and the C-terminal helices of the SK4 pore-forming α subunit. These molecular events can be modulated by the gating molecule PIP_2_, which is positioned in a tight niche formed by the interlobe segment of CaM and intracellular regions of SK4, namely the S4–S5 helices and the HB segment of the proximal carboxyl terminus of the neighboring pore-forming subunit [8]. As evidenced by this recent work from the group of B. Attali, this region could be advantageously targeted to explore allosteric inhibitors of SK4 channels. It remains to be determined whether this interplay mechanism is specific to SK4 channels.

## 4. Materials and Methods

### 4.1. Computer-Based Imaging and Molecular Dynamics Simulation

Full-length 3-D representations of the SK4 channel were generated based upon the 6CNO.pdb and 6CNM.pdb cryo-EM structures reported by Lee and MacKinnon using MODELLER 9v11 [30]. With a pore radius of ≈3.5 Å at the level of the bundle crossing region Val282 in the pore-lining S6 segment [31,32], the 6CNO.pdb structure, obtained in the presence of 2 mM [Ca^2+^], is regarded as being more representative of the channel open state configuration compared to the more populated state I channel structure (6CNN.pdb). MODELLER based homology modeling was used to correct for partly unresolved residues and to add an external loop extending from Arg123 to Pro142 that is missing from the original cryo-EM representation. External loops (Arg123-Pro142) were not, however, corrected in the SK4 model based upon the 6CNM.pdb structure. Fifty structural models were built and optimized to minimize possible violations on the spatial restraints. The ranking of the model structures was based on PDFs (probability density functions) and DOPE (Discrete Optimized Protein Energy) scores. DOPE assesses the energy of the protein of interest, so the model with the lowest DOPE score was kept for structural analyses and molecular dynamic simulations. Molecular dynamics simulations of the 6CNO.pdb based model were performed using NAMD [33] with explicit solvent water molecules. The SK4 structure was embedded in a DPPC bilayer [34] and solvated in a 135 Å × 135 Å × 153 Å cell containing 56,370 TIP3P model water molecules. Altogether the system consisted of 260,213 atoms including 157 K^+^ and 173 Cl^−^ ions to insure electroneutrality at near physiological concentration. An identical procedure was used for the 6CNM.pdb based systems with 71,508 TIP3 type water molecules for a total of 283,036 atoms. Electroneutrality was achieved by the addition of 205 K_+_ and 250 Cl^−^ ions. Cut-on and cut-off parameters needed to define non-bonded interactions were set to 10 Å and 12 Å, respectively, and SHAKE constraints were used to fix lengths of bonds involving hydrogen atoms. Trajectories were generated for a total simulation time of 1.6 µs using a time step of 2 fs for the 6CNO.pdb based models and 0.4 µs for systems based upon the 6CNM.pdb cryo-EM structure. The simulation protocol also includes an equilibration period of 10 ns. Molecular dynamics simulations were performed for a system at constant pressure (1 atm) and constant temperature (300° K). Salt bridge analysis was performed, for 3-D models of the complete channel complex, using the VMD Salt Bridges Plugin, Version 1.0, with oxygen-nitrogen cut-off set at 3.2 Å (Salt Bridges Plugin, Version 1.1 (uiuc.edu)). Significant salt bridge interactions were identified between the C-lobe of CaM and the C-terminal helices of SK4. Contact areas were determined by measuring the loss in solvent-accessible surface area (SASA) for each SK4 residue resulting from the presence of CaM. A similar procedure was used to study contact areas in CaM, by measuring the impact on SASA in the presence of SK4.

### 4.2. Recombinant DNA Techniques 

The human SK4 (GenBankTM accession number AF000972) of hSK4 was subcloned in commercial vectors under the control of the pMT21 promoter as described elsewhere [35,36]. The human CaM (GenBankTM accession number M27319) was subcloned in frame in the pIRES-DsRed vector carrying a Histidine (6xHis) and a cMyc (EQKLISEEDL) tag after the C-terminal [37]. All cDNA mutations were produced with the Q5 Site-Directed Mutagenesis Kit (New England Biolabs Inc., Whitby, ON, Canada), according to the manufacturer’s instructions [38]. All constructs were verified by automated double-stranded sequence analysis (Centre d’expertise et de services Génome Québec, Montreal, QC, Canada). The protein expression at the correct molecular weight was confirmed by standard western-blot analysis for each construct after recombinant expression. HEK293T cells (herein referred to as HEK293T) were grown in Dulbecco’s high-glucose minimum essential medium (DMEM-HG) supplemented with 10% Fetal Bovine Serum, 1% penicillin-streptomycin at 37 °C under 5% CO_2_ atmosphere. HEK293T cells (80% confluence) were transiently transfected using Lipofectamine™ 2000 Transfection Reagent (Life Technologies Inc., Burlington, ON, Canada) using a DNA:lipid ratio of 1:2.5 [38,39]. 

### 4.3. Cell Surface Fractionation Assays

Cell surface fractionation assays were carried out as explained previously [38,39,40,41]. Briefly, transfected HEK293T cells cultured in 100 mm dishes were homogenized at 4 °C in a Tris-based solution containing a mixture of protease inhibitors (Sigma-Aldrich, Oakville, ON, Canada) at pH 7.4. The cell homogenate was aliquoted to collect protein fractions localized in total cell lysates, cytosolic, total membrane, and plasma membrane compartments. In the first step, total cell homogenates were incubated at 4 °C with 1% (*v*/*v*) Triton X-100, before being centrifuged at 10,000× *g* for 10 min to remove cell debris, nuclei, and mitochondria. The pellet was discarded, and the supernatant was kept as the total protein fraction. The cytosolic protein fraction was extracted as the supernatant from the second aliquot after a centrifugation step carried out at 200,000× *g* and 4 °C for 20 min. The pellet of the second aliquot was then resuspended in homogenizing buffer containing 1% (*v*/*v*) Triton X-100. After a 30 min incubation period carried out on ice, an additional centrifugation step was performed at 200,000× *g* to obtain the total membrane protein fraction. Finally, the third aliquot was centrifuged at 10,000× *g* for 10 min at 4 °C. The supernatant obtained was centrifuged at 200,000× *g* and 4 °C for 20 min. The pellet was resuspended in the homogenizing buffer containing 0.6 M KCl. Two centrifugation steps, performed at 200,000× *g* at 4 °C, were required to wash out the KCl. The final pellet, resuspended in the homogenizing buffer, is considered to be enriched in plasma membrane proteins. Proteins (20 μg) were electrophoresed on a 12% SDS-polyacrylamide gel.

### 4.4. Co-Immunoprecipitation Assays

HEK293T cells were transiently transfected with 4 μg pMT21-hSK4 (WT and variants) and 4 μg pIRES-DsRed-HisB-cMyc-CaM (WT and variants). As detailed elsewhere [40], forty-eight hours after transfection, cells were homogenized in 20 mM NaMOPS (pH 7.4), 300 mM NaCl (S7653 Sigma-Aldrich, Oakville, ON, Canada) and 1% digitonin supplemented with EDTA-free protease inhibitors (Thermo Fisher Scientific # 87785). In the absence of any divalent cation chelator, the contaminating Ca^2+^ concentrations were estimated to be hovering in the 10–50 μM range. Homogenates were sonicated, incubated for 1h at 4 °C, and centrifuged at 13,000 rpm for 30 min at 4 °C. A fraction (50–30 μg) of the starting material was set aside as the input fraction and was immunoblotted to confirm protein expression. Co-immunoprecipitation was carried out using 300–200 μg of the homogenates diluted in 150 μL 20 mM NaMOPS (pH 7.4), 300 mM NaCl buffer and mixed by pipetting. The 200 ± 20 μL protein solution was incubated overnight with 50 μL of anti-His magnetic beads (MBL, D291-11). Beads were collected using a PureProteome™ Magnetic Stand (Sigma-Aldrich, Oakville, ON, Canada). The magnetic beads were washed three times with a buffer containing 20 mM NaMOPS (pH 7.4), 300 mM NaCl, and 0.2% digitonin. The bound proteins were eluted with 20 μL of Laemmli buffer at 95 °C for 5 min, electrophoresed on a 12% SDS-polyacrylamide gel, and transferred onto a nitrocellulose membrane. Western blotting was carried out with anti-SK4 (Abcam, ab75956, 1:1000, Cambridge, MA, USA) with an anti-rabbit as secondary antibody (Jackson ImmunoResearch, 1:10,000, Burlington, ON, Canada) and with an anti-His antibody (ThermoFisher 37-2900, 1:1000, Burlington, ON, Canada) with an anti-mouse as secondary antibody (Jackson ImmunoResearch, 1:10,000, Burlington, ON, Canada) to detect the CaM constructs. GAPDH antibody (Sigma-Aldrich, 1:10,000, Oakville, ON, Canada) was used to assess total protein quantity in the input lanes. Signals were detected with the ECL substrate (ThermoFisher, Supersignal™ A38556). Blots were visualized with the ChemiDoc™ Touch Imaging System (Bio-Rad). Molecular weights were estimated using Image Lab software version 5.2 (Bio-Rad) by linear regression of standard molecular weight markers.

### 4.5. Whole-Cell Patch-Clamp Experiments

Patch-clamp experiments were carried out in HEK293T cells with endogenous CaM or with recombinant CaM. In the latter case, HEK293T cells were transiently transfected with 2 μg pIRES-DsRed-HisB-cMyc-CaM (WT or variants) 6 h before the transfection of 1 μg pMT21-hSK4 (WT or variants) to minimize the formation of endogenous CaM-bound channels. For experiments carried out without the fluorescently-tagged red CaM construct, cDNA coding for peGFP (0.2 µg) was included in the cDNA mixture as a marker of successful transfection. The culture medium was changed, and cells were detached with 0.05% trypsin before being replated 24 h post-transfection of pMT21-hSK4 WT or variants. Patch-clamp experiments were carried out 24 h later using the Axon™ Axopatch 200B Microelectrode Amplifier (Molecular Devices). When CaM constructs were expressed, only the cells emitting a red color under the fluorescent microscope were used for the patch-clamp experiments. Electrodes were filled with a solution containing (in mM): 140 KCl; 0.6 NaGTP; 3 MgATP; 10 EGTA; 10 HEPES; titrated to pH 7.4 with NaOH and either 0.1 μM or 0.03 μM free Ca^2+^ (Cai). Ca^2+^ was added from a CaCl_2_ stock solution using concentrations calculated fom the MaxChelator website (UC Davis Health). Pipette resistance ranged from 3 to 4 MΩ. Cells were bathed in a modified Earle’s saline solution (in mM): 145 NMDG; 3 KCl; 10 HEPES; 1CaCl_2_; 2MgCl_2_; titrated to pH 7.4 with NaOH. pClamp™ software 11.2 coupled to a Digidata 1440A acquisition system (Molecular Devices) was used for online data acquisition and analysis. Cellular capacitance was estimated by measuring the time constant of current decay evoked by a 10 mV depolarizing pulse applied to the cell from a holding potential of −100 mV. Currents were elicited by a 3 s voltage ramp applied between +120 and −100 mV from a holding potential of −80 mV at a frequency of 0.2 Hz. TRAM-34 sensitive SK4 currents were estimated from the subtraction of whole-cell current traces obtained after the addition of 0.3 µM TRAM-34 to the bath (Sigma, T6700). The biophysical parameters were routinely measured with SK4 WT and CaM WT as a control for each new experimental condition to validate the transfection efficiency. Average current densities measured at +60 mV are reported in the table. All experimental conditions were tested in “n” different cells from “N” series of independent cultures or transfections and experiments were performed at room temperature, 22–23 °C.

### 4.6. Statistical Analysis

Data was analyzed using a combination of pCLAMP software 11.2, Microsoft Excel, and OriginPro 2020 (OriginLab Corporation, One Roundhouse Plaza, Suite 303, Northampton, MA 01060, USA). Data were expressed as mean ± SD. Statistical significance was determined by Student’s *t*-test in Excel 2016. The level of statistical significance was set at *p* < 0.05.

## 5. Conclusions

Current models of the SK4 gating process rest chiefly upon a large conformational movement of the SK4 S_45_A/S_45_B helices initiated upon the binding of Ca^2+^ to the N-lobe of CaM. CaM binding to the intracellular HA and HB helices of SK4 is considered to remain relatively stable without causing major conformational changes following the initial step. Our current dataset comparing CaM binding carried out in physiological buffers containing contaminating Ca^2+^ ions (supraphysiological micromolar Ca^2+^ concentrations) and channel function measured at physiological intracellular submicromolar Ca^2+^ concentrations, argue that the Ca^2+^-sensitivity of channel activation is not solely predicted from the relative strength of CaM binding onto the HB helix of SK4. Therefore, molecular modifications of the C-terminal appear to influence the channel activation gating. It remains to be seen whether this interpretation can be generalized to other SK channel isoforms. 

## Figures and Tables

**Figure 1 ijms-25-04255-f001:**
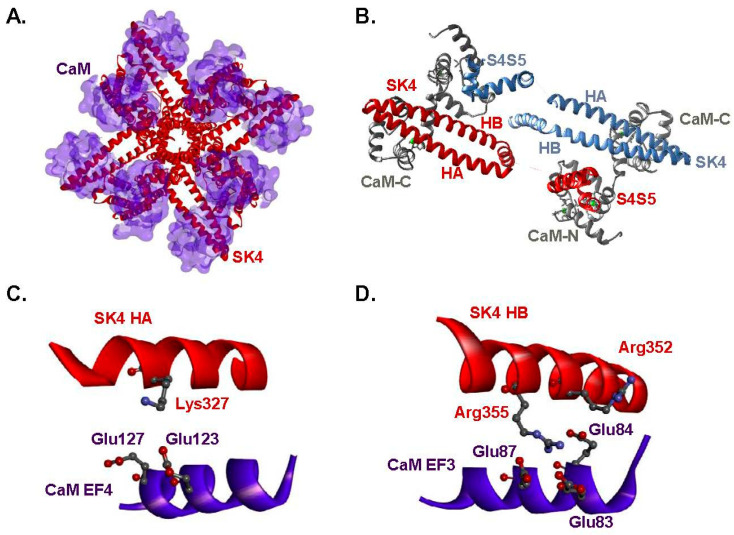
Predicted sets of electrostatic interactions site between CaM and intracellular HA/HB helices of SK4. (**A**). The cryo-EM structure of the tetrameric SK4 channel in complex with four CaM proteins viewed from the intracellular face. This 3-D model was obtained using the Ca^2+^-bound 6CNO.pdb structure. The missing extracellular loops were reconstituted by MODELLER9v11. As seen, a single CaM protein interacts with two channel pore-forming α subunits. (**B**). Details of the intracellular interface between CaM and SK4 viewed from below the cell surface. Only two out of the four pore-forming α subunits are shown for clarity. The region between the S4–S5 linker and the C-terminus was omitted. CaM proteins are depicted in silver whereas the intracellular helices HA and HB for each channel of the two pore-forming α subunit herein depicted are colored in blue and red, respectively. The intracellular HA and HB helices of one channel subunit folds back under the channel in a fashion anti-parallel to the plane of the membrane. The C-lobe of one CaM molecule (CaM-C) is anchored to the HA and HB helices, whereas the N-lobe of the same CaM molecular (CaM-N) forms a complex with the S_45_A helix of a different pore-forming α subunit. Ca^2+^ ions are represented as green spheres. (**C**). Predicted interactions between SK4 HA helix with residues in the CaM C-lobe EF-4. (**D**). Predicted interactions between the SK4 HB helix and the CaM C-lobe EF-3. Original figures were produced with Discovery Studio.

**Figure 2 ijms-25-04255-f002:**
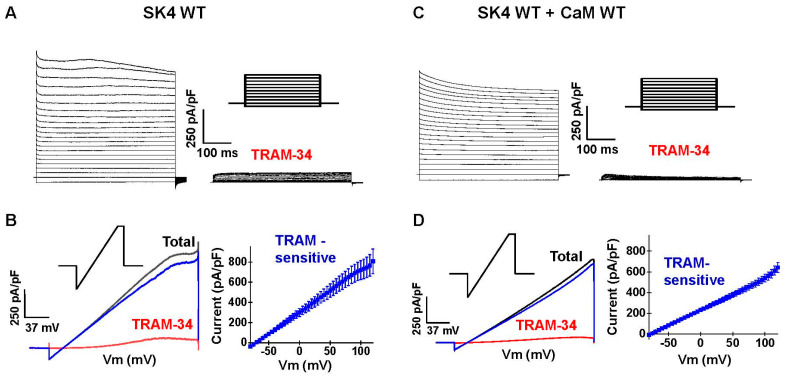
Robust heterologous expression of SK4 in HEK293T cells. pMT21-hSK4 WT was co-transfected without recombinant (endogenous CaM) or with pIRES-DsRed-HisB-cMyc-CaM WT in HEK293T cells. Whole-cell currents were recorded in the presence of 0.1 µM intracellular free Ca^2+^ under a [K^+^ ] gradient of 140 mM (inside)/over 3 mM (out). (**A**). Representative current traces of SK4 WT recorded from HEK293T cells in the absence (left) and presence (right) of 0.3 µM TRAM-34. Currents were activated by 500 ms step voltages between −90 and +120 mV from a holding potential of −80 mV. The voltage pulse protocol is illustrated above the TRAM-34 current traces. (**B**). Left. Representative current traces of SK4 WT recorded using a voltage ramp protocol. Cells were depolarized to +120 mV for 50 ms from a holding potential of −80 mV, followed by a 3-s voltage ramp from +120 to −100 mV before (black) and after (red) the application of TRAM-34. The ramp pulse protocol is illustrated above the total current traces. The TRAM-34 sensitive SK4 WT current (blue) was determined by subtracting the ramp current trace obtained after the addition of 0.3 µM TRAM-34 (red) from the total current (black), and the current densities of TRAM-34 sensitive SK4 WT current were plotted as a function of voltage (Right). (**C**). Representative current traces of SK4 WT co-expressed with CaM WT recorded from HEK293T cells in the absence (**left**) and presence (**right**) of 0.3 µM TRAM-34. (**D**). **Left**. Representative ramp current traces of SK4 WT + CaM WT before (black) and after (red) the application of TRAM-34. The TRAM-34 sensitive SK4 WT + CaM WT current (blue) was estimated by subtracting the current trace obtained after the addition of TRAM-34 (red) from the total current (black) and plotted as a function of applied voltage (**right**).

**Figure 3 ijms-25-04255-f003:**
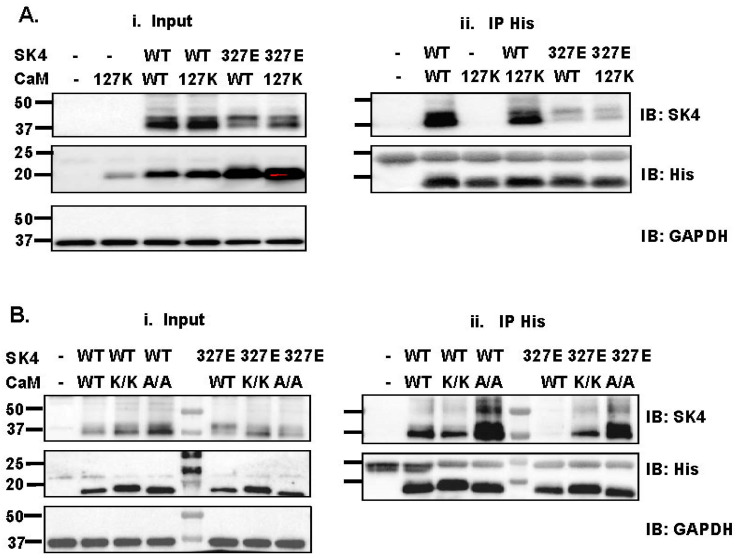
SK4 K327E disrupts SK4/CaM interaction. (**A**). HEK293T cells were transiently transfected with either pMT21-hSK4 WT or K327E combined with either pIRES-DsRed-HisB-cMyc CaM WT or E127K. (**Ai**). All proteins were expressed at the expected molecular mass. (**Aii**). Cell lysates (300 μg) were immunoprecipitated (IP) overnight with anti-His magnetic beads. SK4 WT was pulled down by CaM WT and by CaM E127K. In comparison, the pull-down signal was weaker for SK4 K327E in the presence of CaM WT and CaM E127K (obtained in four different protein lysate preparations during a period of 15 months). (**B**). HEK293T cells were transiently transfected with either pMT21-hSK4 WT, K327E, combined with either pIRES-DsRed-HisB-cMyc CaM WT, E123K/E127K (K/K) or E123A/E127A (A/A). (**Bi**). All proteins were expressed at the expected molecular mass. (**Bii**). Cell lysates (300 μg) were immunoprecipitated (IP) overnight with anti-His magnetic beads. SK4 WT was pulled down by CaM WT, by CaM E123K/E127K, and by CaM E123A/E127A. Pull-down signal was reduced for SK4 K327E + CaM WT but the protein signal was present for the channel pair SK4 K327E + CaM E123K/E127K and the pair SK4 K327E + CaM E123A/E127A confirming that the charge at position Glu123 and Glu127 in CaM may not be a limiting factor in the protein–protein interaction. Similar results were obtained for three different protein lysate preparations during a period of 2 months.

**Figure 4 ijms-25-04255-f004:**
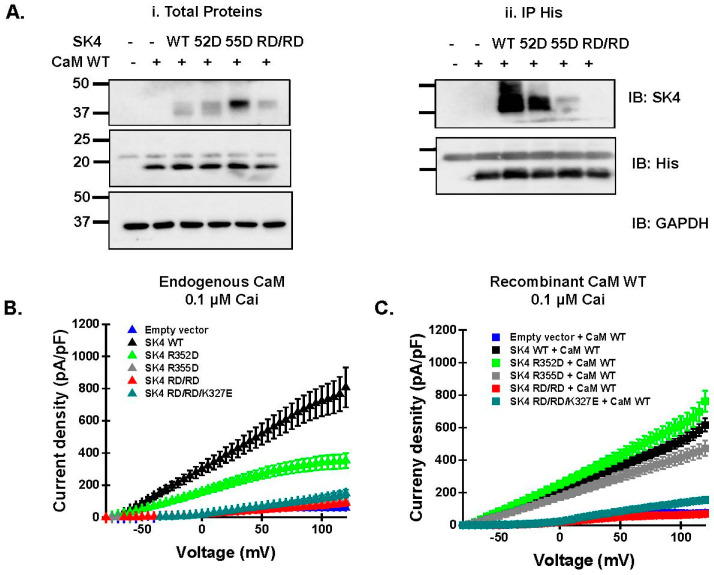
Double substituted channel SK4 R3552D/R355D prevents pull-down by CaM WT. (**A**). HEK293T cells were transiently transfected with either pMT21-hSK4 WT, R352D (52D), R355D (55D), or R352D/R355D (RD/RD) and pIRES-DsRed-HisB-cMyc CaM WT. Cell lysates (300 μg) were immunoprecipitated (IP) overnight with anti-cHis magnetic beads to capture His-tagged CaM. Immunoblotting was carried out after eluting the protein complexes from the beads with anti-SK4 and anti-His antibodies, the latter identifying recombinant CaM proteins. As seen in the (**Ai**) panel, total proteins or input fraction, proteins were expressed at the expected molecular mass. (**Aii**). The protein signal remained strong for SK4 WT after protein elution from the anti-His tagged beads, but signals are considerably weaker for SK4 R352D and near eliminated for SK4 R355D and SK4 R352D/R355D, suggesting the loss of a strong protein interaction. Similar results were obtained for 8 different protein lysate preparations during a period of 36 months. (**B**). pMT21-hSK4 WT, R352D, R355D or R352D/R355D were transfected in HEK293T cells. Experiments were also carried out in HEK293T cells transfected with the empty pMT21 vector (shown in blue). TRAM-34 sensitive currents were recorded in the presence of 100 nM intracellular free Ca^2+^ with a [K^+^] gradient of 140 mM (inside)/3 mM (out) over the −80 mV to +120 mV range. Average current densities are shown ± SEM. Current densities recorded with SK4 R355D (gray) or R352D/R355D (red) were indistinguishable from currents measured in mock-transfected cells (blue). (**C**). Empty pMT21 vector, pMT21-hSK4 WT, R352D, R355D or R352D/R355D were co-transfected with pIRES-DsRed-HisB-cMyc CaM WT in HEK293T cells. TRAM-34 sensitive currents were recorded under the same conditions as in (**C**). Overexpression of CaM WT did not improve the SK4 R352D/R355D current density, but significantly increased the current density of SK4 R355D. Average current densities ± SD recorded at +60 mV, including the statistical analysis, are shown in Table 1.

**Figure 5 ijms-25-04255-f005:**
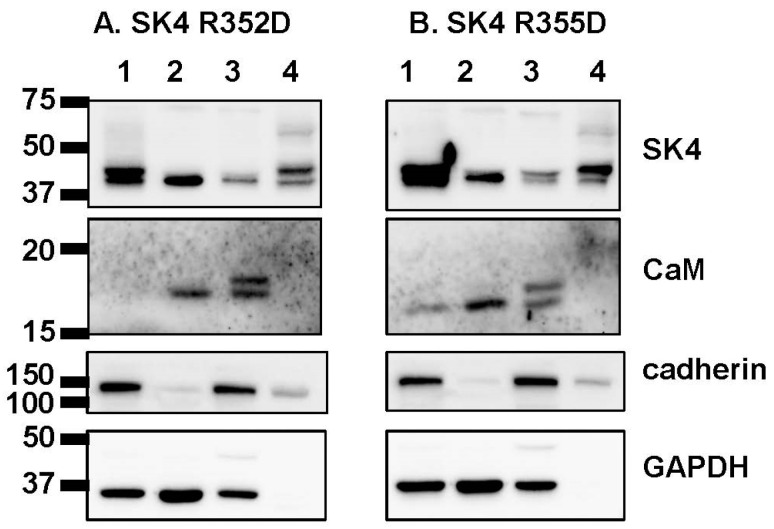
SK4 R352D and SK4 R355D are targeted to the cell surface. HEK293T cells were transiently transfected with SK4 WT and SK4 R352D (Panel **A**), or SK4 WT and SK4 R355D (Panel **B**). Two days after transfection, the cells were lysed, and cell fractions were obtained through preparative ultracentrifugation as described in the Experimental procedures section. Western blotting was carried out for the four protein fractions identified as such: lane 1, total proteins; lane 2, cytoplasmic proteins; lane 3, total membrane proteins; and lane 4, enriched plasma membrane proteins. The proteins were probed with the following antibodies: SK4 (Abcam; ab75956, 1:1000 dilution); CaM (Millipore; catalog no.: 05-193, 1:1000 dilution) with anti-mouse (1:10,000 dilution); and cadherin (Pan-cadherin; Thermo Fisher; catalog no.: 71-7100, 1:1000 dilution) with anti-rabbit (1:10,000). Cadherin was used as a marker for the plasma membrane. Each lane was loaded with 20 μg proteins. The molecular markers are shown to the left of the blots with the value provided in kilodalton. The molecular masses were estimated by linear regression and interpolation from the molecular mass markers using the Image Lab software, version 5.2 (Bio-Rad Laboratories (Canada) Ltd, Saint-Laurent, QC H4R 2E).

**Figure 6 ijms-25-04255-f006:**
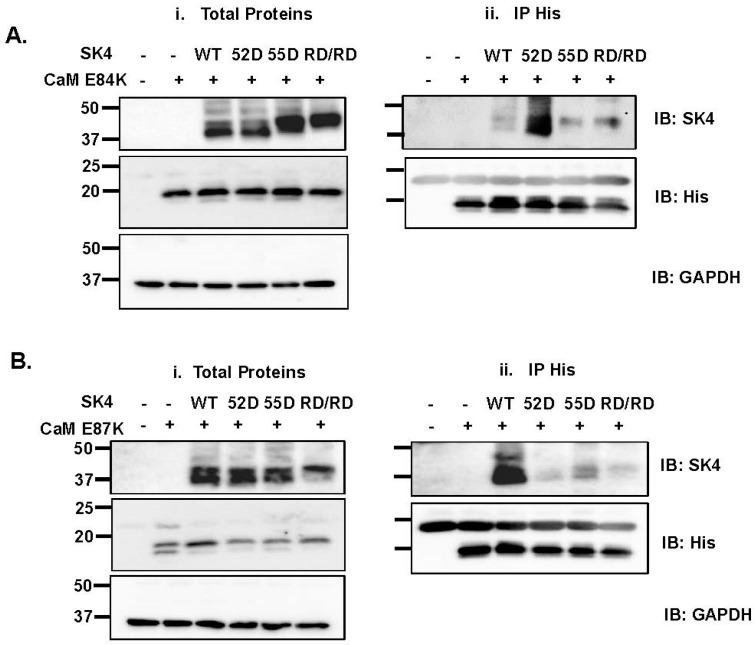
CaM E84K disrupts the SK4/CaM interaction. (**A**). HEK293T cells were transiently transfected with pMT21-hSK4 WT, R352D (52D), R355D (55D), or R352D/R355D (RD/RD) and pIRES-DsRed-HisB-cMyc CaM E84K. Cell lysates (300 μg) were immunoprecipitated (IP) overnight with anti-His magnetic beads as described above. (**Ai**). All proteins were expressed at the expected molecular mass. (**Aii**). SK4 WT, R355D, and R352D/R355D were not pulled down by CaM E84K but a clear protein signal was observed for R352D. These results indicate that a negatively charged residue at CaM 84 is required to carry constitutive interaction of CaM with SK4 WT and that this interaction could involve a positively charged residue at position 352 in SK4. Similar results were obtained for 3 different protein lysate preparations during a period of 24 months. (**B**). HEK293T cells were transiently transfected with pMT21-hSK4 WT, R352D (52D), R355D (55D), or R352D/R355D (RD/RD) and pIRES-DsRed-HisB-cMyc CaM WT or E87K. (**Bi**). All proteins were expressed at the expected molecular mass. (**Bii**). Cell lysates (300 μg) were immunoprecipitated (IP) overnight with anti-His magnetic beads, as described above. SK4 WT was pulled down by CaM E87K but SK4 R352D, SK4 R355D, and SK4 R352D/R355D were not co-immunoprecipitated. Similar results were obtained for 3 different protein lysate preparations during a period of 24 months.

**Figure 7 ijms-25-04255-f007:**
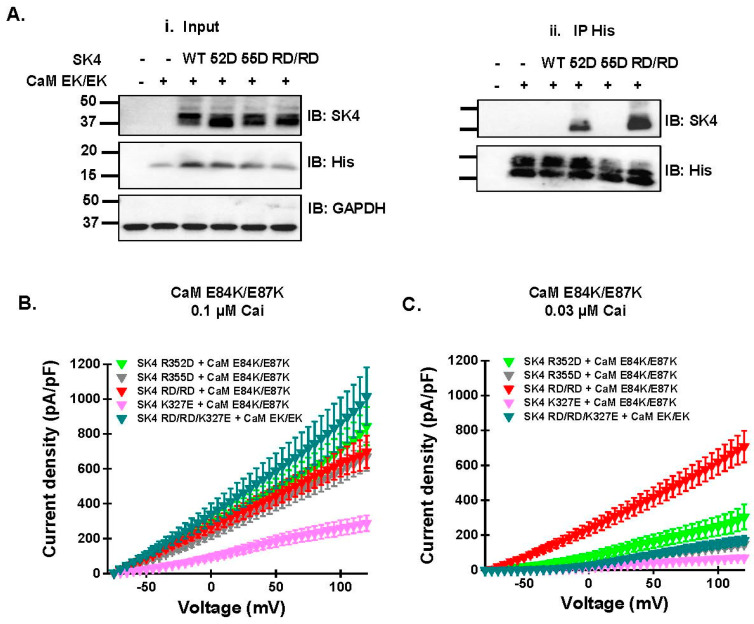
Charge reversal in both SK4 and CaM rescued interaction and function. (**A**). HEK293T cells were transiently transfected with pMT21-hSK4 WT, R352D (52D), R355D (55D), or R352D/R355D (RD/RD) and pIRES-DsRed-HisB-cMyc CaM WT or CaM E84K/E87K (EK/EK). (**Ai**). All proteins were expressed at the expected molecular mass. (**Aii**). Cell lysates (300 μg) were immunoprecipitated (IP) overnight with anti-His magnetic beads as described above. SK4 WT was not pulled down by CaM E84K/E87K but SK4 R352D and SK4 R352D/R355D were successfully co-immunoprecipitated. Similar results were obtained for 3 different protein lysate preparations during a period of 24 months. (**B**). TRAM-34 sensitive currents for SK4 WT, R352D, R355D, R352D/R355D, K327E, and R352D/R355D/K327E were measured at 0.1 µM Cai following co-expression with CaM E84K/E87K. Average current densities are shown ± SEM. Complete set of average current densities ± SD recorded at +60 mV, including the statistical analysis, are shown in Table 2. (**C**). TRAM-34 sensitive currents for SK4 WT, R352D, R355D, R352D/R355D, K327E, and R352D/R355D/K327E were measured at 0.03 µM Cai following co-expression with CaM E84K/E87K. Average current densities are shown ± SEM. Complete set of average current densities ± SD recorded at +60 mV, including the statistical analysis, are shown in Table 2.

**Table 1 ijms-25-04255-t001:** Whole-cell K^+^ currents were recorded from HEK293T cells transiently transfected with pMT21-hSK4 WT or variants with or without pIRES-DsRed-HisB-cMyc CaM WT and variants in the presence of 0.1 μM Cai^+^ and a 3 mM/140 mM outward gradient for K ions. Qualitative co-immunoprecipitation signal intensities (Co-IP) for SK4 substituted proteins are reported relative to SK4 WT with “++” meaning that the signal intensity was equal or superior to SK4 WT; “+” meaning that the signal intensity was detected but weaker than for WT; and “–“ meaning that the protein signal could not be detected under the same exposure conditions. Current densities of TRAM-34 sensitive currents are reported at a voltage of +60 mV. The n/N value refers to the number “n” of independent recordings obtained in “N” different series of transfections. SK4 R352D/R355D is referred to RD/RD. Data were expressed as mean ± SD. Statistical significance was determined by Student’s *t*-test in Excel. The level of statistical significance was set at *p* < 0.05 and was estimated against (*v.*) the identified constructs.

SK4	Co-IP	Native CaM		CaM WT	
		n/N	Mean ± SD (pA/pF)	n/N	Mean ± SD (pA/pF)
Empty vector	-	6/2	55 ± 16	5/1	77 ± 20
WT	++	10/2	563 ± 199 *p* < 0.001 *v*. Empty vector	33/9	401 ± 135 *p* < 0.001 *v.* Empty vector + CaM WT *p* = 0.03 *v.* SK4 WT + native CaM
K327E	+	14/2	183 ± 50 *p* < 0.001 *v.* Empty vector *p* < 0.001 *v*. SK4 WT + native CaM	27/5	405 ± 193 *p* < 0.001 *v.* Empty vector *p* < 0.001 *v*. SK4 K327E + native CaM
R352D	+	18/2	289 ± 148 *p* < 0.001 *v.* Empty vector *p* = 0.001 *v*. SK4 WT + native CaM	21/4	430 ± 163 *p* < 0.001 *v.* Empty vector + CaM WT *p* = 0.008 *v.* SK4 R352D + native CaM
R355D	+	20/3	59 ± 23 *p* < 0.001 *v.* SK4 WT + native CaM	37/5	330 ± 172 *p* < 0.001 *v.* Empty vector + CaM WT *p* < 0.001 *v.* SK4 R355D + native CaM
RD/RD	-	15/3	49 ± 22 *p* < 0.001 *v.* SK4 WT + native CaM	11/2	54 ± 19 *p* < 0.001 *v.* SK4 WT + CaM WT
RD/RD/K327E	N/A	9/1	75 ± 30 *p* < 0.001 *v.* SK4 WT + native CaM	10/1	97 ± 31 *p* < 0.001 *v.* SK4 WT + CaM WT

**Table 2 ijms-25-04255-t002:** Whole-cell K^+^ currents were recorded from HEK293T cells transiently transfected with pMT21-hSK4 variants with pIRES-DsRed-HisB-cMyc CaM E84K/E87K in the presence of a 3 mM/140 mM outward gradient for K^+^ ions and either 0.1 μM intracellular Ca^2+^ or 0.03 μM intracellular Ca^2+^. Qualitative co-immunoprecipitation signal intensities (Co-IP) for SK4 substituted proteins are reported relative to SK4 RD/RD with “++” meaning that the signal intensity was equal or superior to SK4 RD/RD and “–“ meaning that the protein signal could not be detected under the same exposure conditions. Note that SK4 WT could not be pulled down by CaM E84K/E87K. Current densities of TRAM-34 sensitive currents are reported at a voltage of +60 mV. The n/N value refers to the number “n” of independent recordings obtained in “N” different series of transfections. SK4 R352D/R355D is referred to RD/RD. Data were expressed as mean ± SD. Statistical significance was determined by Student’s *t*-test in Excel. The level of statistical significance was set at *p* < 0.05 and was estimated against (*v.*) the identified constructs.

SK4	CaM	Co-IP	Cai 0.1 μM		Cai 0.03 μM	
			n/N	Mean ± SD (pA/pF)	n/N	Mean ± SD (pA/pF)
R352D	E84K/E87K	++	19/3	525 ± 172	8/1	182 ± 73 *p* < 0.001 *v*. Cai 0.1 μM
R355D	E84K/E87K	–	26/3	468 ± 136	12/2	92 ± 30 *p* < 0.001 *v*. Cai 0.1 μM
RD/RD	E84K/E87K	++	26/5	511 ± 205	10/2	457 ± 187
K327E	E84K/E87K	N/A	7/1	202 ± 61	7/1	48 ± 13 *p* < 0.001 *v*. Cai 0.1 μM
RD/RD/K327E	E84K/E87K	N/A	8/1	642 ± 275	6/1	106 ± 42 *p* < 0.001 *v*. Cai 0.1 μM

## Data Availability

The 3-D model and M.D. simulations will be made available upon request to R.S. All other data are included in the article.

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
