# Peer review of "Investigating the Impact of Electrostatic Interactions on Calmodulin Binding and Ca^2+^-Dependent Activation of the Calcium-Gated Potassium SK4 Channel"

_ijms, 2024, doi:10.3390/ijms25084255_

Round 1

Reviewer 1 Report

Comments and Suggestions for Authors

In this manuscript Segura et al investigate how specific charged amino acid residues in the C-terminus of SK4 potassium channel contribute to constitutive binding of calmodulin and Ca2+ dependent gating. Using the molecular dynamics simulations, it is hypothesized that SK4 Lys327, Arg352 and Arg355, as well as CaM Glu84,  Glu87 and Glu127 form salt bridges between SK4 and CaM. The most striking results are that R352D/R355D SK4 double mutant loses its’ ability to bind CaM in pull-down experiments or produce K+ currents in patch clamping, but co-expression of SK4 R352D/R355D with CaM E84K/E87K double mutant restores both the co-immunoprecipitation and the K+ currents, presumably due to restoration of the salt bridges between the positive and negative charges on CaM and SK4. The results of the single substitutions at these positions are not that straight forward, suggesting complex interactions at these sites.  Overall, the paper presents interesting and valuable results, however, it lacks clarity and is difficult to follow.

Specific comments (both major and minor):

1. line 25. “for constitutive binding of CaM binding to the intracellular HA and HB”

2. line 26. “the charge substitution R355D” – charge reversal

3. line 30. “Overexpression of CaM WT rescued the function of SK4 R352D, R355D, and K327E to the current density of SK4 WT” . Add -  “, but not the function of SK4 R352D/R355D”.

4. Lines 31, 32. “Co-expressing SK4 R352D/R355D with CaM E84K/E87K, predicted to face the HB helix in SK4, nonetheless restored pull-down to CaM and normalized whole-cell potassium current density at 0.03 μM Cai.”   Start with “However” and delete “nonetheless”.

5. Line 57-60. “Each lobe can bind up to two Ca2+ ions as the glutamate residues contribute their oxygen atoms to its two distinct EF hands… Each EF hand motif can bind one Ca2+ ion and alanine substitution of the aspartate residue … abolishes Ca2+ binding.” These two sentences are poorly connected. What’s important? Glutamates, aspartates, or both?

6. Line 97. “…and Glu127) from CaM”. Delete )

7. Line 108. “… co-immunoprecipitation of the full-length SK4 and CaM (WT and substituted proteins) carried out in nominally Ca2+ -free solutions…” Nominally Ca2+ free NaCl based solutions without addition of EGTA, or other Ca2+ chelates, contain between 20 and 50 µM Ca2+. From physiological point of view, this is a saturating concentration of Ca2+, so in this case “Ca2+ free” is misleading, and the authors should either measure Ca2+ content in their solutions, or explicitly say that “nominally Ca2+ free” means “high Ca2+”.

8. Lines 116-121. To understand the meaning of what’s written here, one should read the results first. So, this paragraph should be simplified to level suitable for the introduction.  

9. Line 122.(also 532) “…despite the seemingly absence…”.  Should be “apparent absence” or “seeming absence”

10. Line 144. “…emphasizing significant contributions from SK4 Arg352 and Arg355 as well as CaM Glu84 and Glu127.” Do you mean Glu127, or Glu87?

11. Line 151. “whole-cell configureuration”

12. HEKT cells. The standard name for these cells HEK293T.

13. Table 1. K327E;  183 ± 50. How this compares to the empty vector and WT SK4?

14. Figure 6, legend, lines 326-332. All pull-down data should be presented in a Table (including listed in this figure), similarly to the K+ current data. It would be ever better, if for each pair of constructs there were both, their interaction status from pull-downs and their function from patch clamping, in one table.

15. The rationale, and meaning of the differences between the patch clamping experiments with 0.1 µM and 0.03 µM Ca2+ should be better explained.  

16. Line 413.  CAMBD. The standard abbreviation is CaMBD.

17. Discussion. Lines 426-442. This part of the discussion is not directly related to the results and can be removed. The discussion should start with the summary of the results.

18. Line 450 “…pull-down of Ca2+-free CaM to SK4 WT”.  - Pull-down of SK4 WT by Ca2+-free CaM. Also Line 453 “CaM pull-down to SK4”. It’s either “by SK4” or “of SK4”.

19. Lines 459, 462. Delete second “nonetheless”

20. Discussion, section 3.2. Text between lines 498-517 can be deleted.

21. Conclusions. Lines 678-679. How this statement is connected to the presented data is far from obvious.

Comments on the Quality of English Language

Some editing for better expression and clarity is required

Reviewer 2 Report

Comments and Suggestions for Authors

“Investigating the impact of high-affinity constitutive calmodulin binding on the Ca2+ dependent activation of the calcium gated K SK4 channel”

In this manuscript, the authors investigated the interaction between SK4 and CaM. They identified a salt bridge between 2 Arg residues in SK4 and 2 Glu residues in CaM and show that this interaction is required for constitutive CaM binding to SK4. Interestingly, the Ca2+ sensitivity of SK4 gating process was influenced by electrostatic network between CaM C-lobe and SK4 HB helix.

The strength of CaM-SK4 interaction was assessed by co-immunoprecipitation, using WT and substituted proteins, and the activity of CaM-SK4 was assessed by whole cell patch-clamp recordings.

This is an interesting article with data providing new insight into the role of CaM-SK4 constitutive interaction in channel gating.

Here are few questions and suggestions:

For the different IP, it could be indicated that they are representative of N experiments. In the legends of these figures it is written “Positive pull-down signals were obtained for SK4 with …” it is a bit confusing because it referred to data that is shown and data that is not shown on the figures (Lines 213-216; 262-268). If supplemental data is available it would be worth showing as a supplement and mentioning in the figure legends only the information about data that is shown.

About figure 3:

The expression level of CaM E127K appeared impaired in absence of SK4 expression (3A). Would that suggest that SK4 enhanced CaM expression?

About tables showing whole cell currents: instead of a table, I would have preferred a histogram with dots showing the different currents registered. This presentation would give information about current variability and it would be easier to compare the different combinations. Adding to this figure a line with IP results (CaM-SK4 interaction maintained or not) would be very helpful.

Absence of Co-IP or reduced signal for Co-IP: do you think there might be still some interaction between few CaM and SK4 that are not detected and that would explain remaining currents? In other words, the absence of SK4 immunodetection in the different IP would not mean absence of interaction but faint interaction.

For curiosity, did you simulate the behavior of the SK4 HB helix with CaM C-lobe when Arg352 is substituted by His, the SK4 mutations that was associated with dehydrated hereditary xerocytosis? This R352H substitution modified the Ca2+ sensitivity of the channel as well as the maximal current amplitude.

The rectification of SK4 current seemed to disappear when CaM is overexpressed. Same observation with expression of CaME84K/E87K. Do you have any interpretation of this observation?

About figure 5: what is the band appearing in lane 3 (membrane) with CaM immunodetection? In the other WB, a non-specific band around 20 kDa was present but here, the band appearing in lane 3 is largely below 20 kDa. Why the CaM is not detected in enriched membrane (lane 4 with more SK4 signal) whereas it is still present in membrane fractions (lane 3 with less SK4 signal)?

Line 151 misspelling for “conFigureuration”

Round 2

Reviewer 1 Report

Comments and Suggestions for Authors

The authors have adequately responded to my comments, improving the paper considerably.